# Influence of Local Factors on Coastal Erosion: The Case of Vagueira Beach in Portugal

**Luiz Magalhães Filho** [1,2,*], **Peter Roebeling** [1] and **Carlos Coelho** [3]

1 CESAM & Department of Environment & Planning (DAO), University of Aveiro (UA), 3810-193 Aveiro, Portugal

2 Federal Institute of Education, Science and Technology of Tocantins (IFTO), Dianopolis 77300-000, Brazil

3 RISCO & Department of Civil Engineering (DECivil), University of Aveiro (UA), 3810-193 Aveiro, Portugal

* Correspondence: luizlacerda@ua.pt

**Abstract:** Vagueira Beach, on the central Portuguese coast, is known as one of the places in Europe most affected by coastal erosion. The area has suffered more than 156 m of coastline retreat from the period 1958 to 2001. With the aim of evaluating the influence of local factors on coastal erosion, this paper assesses the anthropogenic and natural factors that are related to the retreat of the coastline by adopting statistical correlation and regression analyses. Through Pearson's correlation coefficient ($r$), it was observed that local factors, such as annual dredging at the Aveiro Port entrance ($r = 0.93$), the total length of groins in the Espinho–Vagueira section ($r = 0.89$), and storm events ($r = 0.52$), are directly related to coastline retreat in the area. A multiple linear regression model was developed in which coastline retreat is explained by these same factors over the period 1980–2006. With a coefficient determination of $R^2 = 0.91$, it was observed that the length of groins (significant at the 1% level), the dredging of the port entrance (significant at the 5% level), and precipitation (as a proxy for storm events; significant at the 10% level) are significantly correlated with coastline retreat. Hence, it is shown that anthropogenic factors are the main drivers of coastline retreat in Vagueira Beach. This study provides an innovative approach for the assessment of coastal erosion, resulting in important information that could be used for decision-making related to coastal zone management as it allows us to understand in greater detail the main drivers of coastal erosion.

**Keywords:** groins; dredging; storm events; natural factors; anthropogenic factors; regression model

## 1. Introduction

Coastal areas are among the most critical regions for humanity. Indeed, more than 30% of the world's population lives in the coastal zone—which is twice as densely populated as inland areas [1–3]—with the majority of the total population of more than half of coastal countries living within 100 km from the coastline [4]. Coastal zones host the majority of centers of political decision, economic and technical cooperation, as well as a large part of the industries and economic activities in many countries [5]. Hence, the increase in coastal erosion directly threatens the majority of the world's population and economy [6,7].

The coastal zone is one of the most dynamic environments on the planet and coastline position constantly changes at various times and spatial scales [6,8,9]. The position of the coastline is affected by many factors, some due to natural causes (related to coastal dynamics, climate, and climate change) and others due to human interventions (such as urbanization, dredging, and infrastructure) [7,10–12]. As a result of the interaction between these factors, the coastline can move out to sea, stay in place, or be pushed-back toward the continent [10,13,14]. This latter indentation, in turn, causes losses in territory and is considered a process of coastal erosion.

Portugal is located on the Iberian Peninsula in southwest Europe, facing the Atlantic coast. The Portuguese coast is a highly energetic region, suffering from storms generated in

the North Atlantic [15]. Central Portugal is one of Europe's coastal areas that suffer most from the processes of coastal erosion [7] due to the reduction in the delivery of sediments to the coast, rising sea levels, the increase in the frequency of storms, and changes in human settlements [13,16]. Several studies have shown that the Espinho–Mira section is one of the areas most vulnerable to coastal erosion [8,15,17–22]. This area has a variety of beaches, including Vagueira Beach—a small Portuguese village situated in the district of Aveiro that annually receives a large inflow of tourists. Vagueira Beach has suffered about 160 m of coastline retreat over the period 1948–2001 [7].

With the purpose of understanding what is aggravating coastal erosion in Central Portugal, the objective of this study is to assess the key anthropogenic and natural factors that influence the rate of coastline retreat. To this end, correlation and regression analyses are performed, relating coastline retreat to anthropogenic and natural factors. This study contributes to previous studies, which have mostly used monitoring and modeling approaches to assess the causes and impacts of coastal erosion [8,9,16,21–23]. Hence, this study developed a correlation and regression analysis to assess the key anthropogenic and natural factors that influence the rate of coastline retreat and, thus, could contribute to the development of coastal erosion adaptation strategies in Central Portugal.

The remainder of this paper is structured as follows. Section 2 (Materials and methods) provides the definition and sources of data, a characterization of the study area, and a description of the statistical analyses performed. In Section 3 (Results), results from the correlation and regression analyses are presented, and the influence of the various anthropogenic and natural factors on coastline retreats are discussed. In Section 4 (Discussion), results are discussed and compared to other relevant studies. Finally, concluding remarks and observations are provided in Section 5.

## 2. Materials and Methods

### 2.1. Characterization of the Study Area

Vagueira Beach is located on the central Portuguese coast, between Costa Nova and Areão Beaches (see Figure 1). This region is influenced by strong urbanization pressure, widespread sedimentary deficit, and frequent wave action and inundation due to energetic storm events [20]. This coastal stretch, marked by the presence of extensive and fragile dunes, sandy beaches of low elevation, and developments parallel to the Ria de Aveiro lagoon, is considered one of the most dynamic types of coasts [8,22].

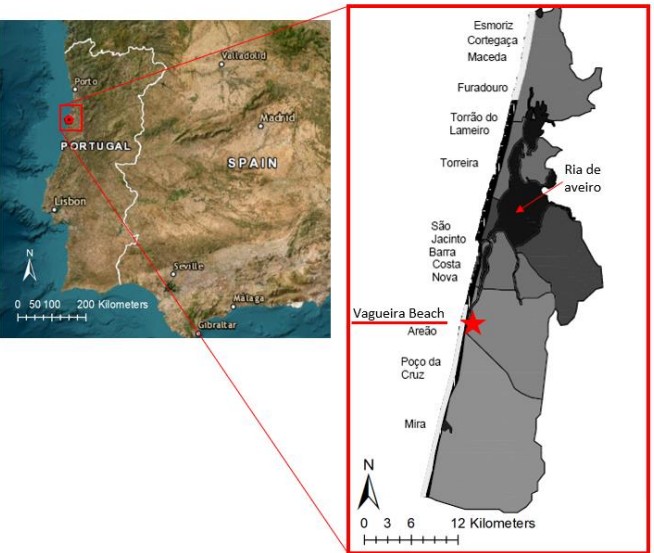

**Figure 1.** Vagueira Beach study area location in the Ria de Aveiro region (adapted from: [21]).

Vagueira Beach shows an increase in coastal erosion—with rates of coastline retreat increasing from 1.6 m/year during the late 1950s to 7.1 m/year over the period 1996–2001 (see Table 1). Note that rates of coastal erosion are grouped into four different periods of time due to the scarcity of specific information, and thus, coastline retreat data are based on the best available information.

**Table 1.** Average rates of the coastline retreat in Vagueira Beach over the period 1958–2001 (source: [7]).

| Period | Average Rate of Coastline Retreat | |
| (years) | Annual (m/year) | Total (m) |
| --- | --- | --- |
| 1958–1969 (12 years) | 1.6 | 19.2 |
| 1970–1979 (10 years) | 2.4 | 24.0 |
| 1980–1990 (11 years) | 3.9 | 42.9 |
| 1991–1995 (5 years) | 5.5 | 27.5 |
| 1996–2001 (6 years) | 7.1 | 42.6 |
| Total | - | 156.2 |

In case the sediment available for littoral drift is equal to the potential sediment transport capacity, the coastline would be in dynamic equilibrium, and the sediment transport along this coastal stretch would forward the sediment's downdrift [22]. The morphology of the sector is mainly defined by wave actions that are responsible for the sediment transport that occurs on the littoral. The central Portuguese coast is exposed to very energetic wave climates, which promote a north-to-south directed net littoral drift, estimated to be around $1.1 \times 10^6$ m$^3$/year [24]. In the past, the Douro River would be able to practically alone provide the amount of sediment needed to balance sediment losses [25]. However, with the construction of dams (between 1972 and 1985), this supply of sediment decreased dramatically [11,26].

Another factor related to coastline retreat is the dredging that occurs at the Aveiro Port entrance, in which part of this material was, in the past, commercialized for other uses (such as construction and industrial use)—further reducing sediment supply from the north and, thus, accentuating the coastline retreat of the Vagueira Beach [22]. Only after 2000 were dredging operations performed for navigation purposes at Aveiro port use in beach and landfill nourishment (when the sediments present the required quality), with only a part deposited at sea, to the south of the breakwater, in an attempt to mitigate coastal erosion.

Anthropogenic interventions and impacts can be even more severe due to the destruction of natural coastal defense structures, in particular the beach and the frontal dune system, which act as the first natural barriers to wave action. Beach tourism, the advance of urbanization, and the construction of industrial areas are the main activities that have resulted in the creation of wind runners, changes in the floodplain, and the extraction of sediments [27]. The rate of urbanization in the Ria de Aveiro region increased from 5% in 1975 to 12% in 2006 [11]. This occupation of the coastal area led to the destruction of dunes that provided large volumes of sand for the dynamic interaction with and natural defense against the sea [25].

Due to the increase in coastal erosion, the vulnerability of human settlements to losses and damages has increased. This has resulted in the construction of heavy engineering structures (groins, rocky revetments, dikes, and breakwaters) to protect these settlements against the sea. The main structures found along the central Portuguese coast are groins. Groins are short structures placed perpendicular to the coastline and extended to the surf zone, usually built in straight portions of the shoreline. Their main purpose is to retain sand and promote updrift accretion [28]. Groins do not add sediment to the coastal system, and secondary negative effects of these structures may anticipate sediment deficit downdrift (see Figure 2). At Vagueira Beach, the main negative impacts are registered due to the northern structures, and this is explained by a predominance of currents from the northwest along the considered coastal stretch [29]. In addition, the construction of the breakwater at

the Aveiro Port entrance in 1942 resulted in the trapping of sediments from the northern part of the considered coastal stretch [30].

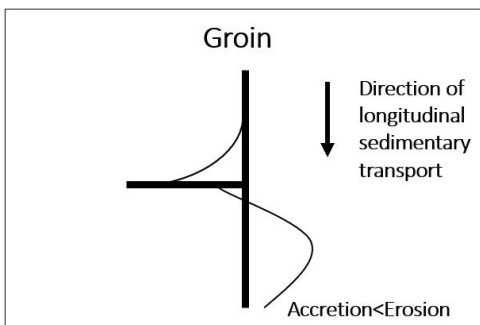 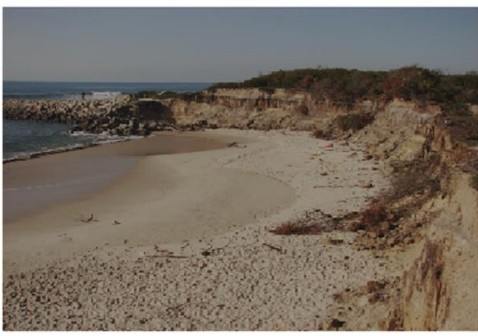

**Figure 2.** Scheme of the consequences resulting from the construction of groins on the coast and effects of erosion immediately south of the groin in Cortegaça Beach, Portugal (adapted from: [29]).

The number of groins built between Esmoriz Beach and Vagueira Beach increased from none in 1970 to nine in 2006. In 2006 there were nine groins of various lengths (between 100 and 200 m each) in this section—totaling 1460 m of groins (see Figure 3). Built with the objective of protecting the areas north of where they are located, they tend to further aggravate the coastline retreat in Vagueira Beach. Note that groin sand retention, accretion updrift, and, in particular, sand depletion and erosion downdrift takes several decades before a stable situation is attained [28], and hence, the impacts of groins are observed until decades after their construction.

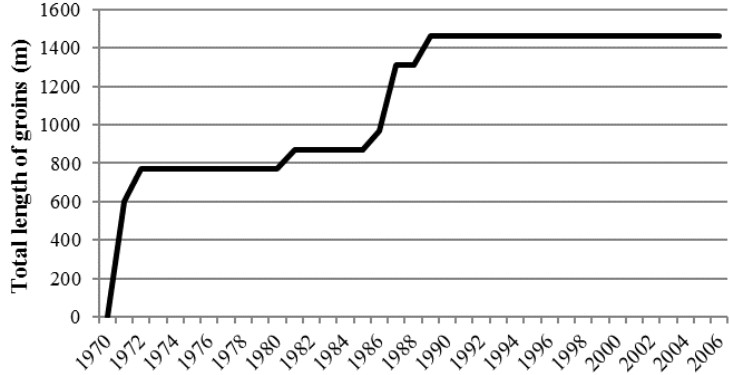

**Figure 3.** Total length of groins built in the Espinho–Vagueira section over the period 1970–2006 (based on: [7]).

Storm events contribute to coastline retreat, changing the beach morphology, including responses to the direct action of waves, winds, tides, and surges, as well as seasonal changes in the surf zone between storms and calm periods [31]. Storms generated in the North Atlantic are frequent in winter and can persist for up to 5 days, with significant wave heights as high as 8 m [31]. However, wave records during storm events are unreliable (incomplete), as during these events, measuring equipment frequently fails due to the high wave energy. A proxy for storm events is precipitation, as periods of intense rainfall (mainly during autumn and winter) are generally accompanied by storm events (Figure 4; see also Section 3.1). Over the period 1980–2010, the median annual precipitation was about 950 mm, with a standard deviation in annual precipitation of about 175 mm.

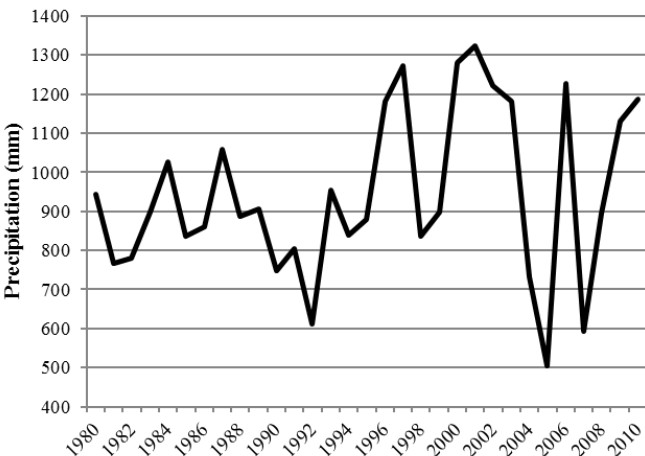

**Figure 4.** Annual precipitation in Aveiro district over the period 1981–2011 (based on: [32]).

*2.2. Data Collection*

To allow for the assessment of the key anthropogenic and natural factors that explain the rate of coastline retreat, corresponding time series data were collected from government bodies and publications for the Espinho–Mira case study area. t = This analysis adopted data over the period of 1980–2001 (21 years), for which information on all considered variables were available. In particular:

(a) Annual coastline retreat (in meters/year) in Vagueira Beach for the period 1980–2006 (obtained from the EUROSION project) [7];

(b) Annual dredging (in m$^3$/year) at the Aveiro Port entrance for the period 1980–2010 (data provided by the Port of Aveiro);

(c) Total length of groins built (in meters) in the Espinho–Vagueira section for the period 1980–2001 (obtained from the EUROSION project) [7];

(d) Annual precipitation (in mm/year) in the Aveiro district for the period 1980–2010 as a proxy for storm events (obtained from the Instituto Português do Mar e da Atmosfera) [32].

*2.3. Statistical Analysis*

In seeking to determine which anthropogenic and natural factors influence coastal erosion in Vagueira Beach, a series of correlation and regression analyses were performed with the statistical software SPSS (Statistical Package for Social Science; version 15.0).

Correlation analysis was applied to assess the correlation between the independent variables (dredging, length of groins, annual precipitation) and the dependent variable (coastline retreat). The coefficient of correlation (*r*) was a measure of the degree of the linear relationship between two quantitative variables (where −1 indicates that there is a perfect negative linear relationship, 0 indicates that there is no linear relationship, and +1 indicates a perfect positive linear relationship). The closer the correlation coefficient is to −1 or +1, the stronger the linear association between the two variables is [33,34]. Pearson's correlation coefficient (*r*) is given by:

$$r = \frac{\sum (x_i - x) \times (y_i - y)}{\sqrt{\left(\sum (x_{i-\bar{x}})^2\right) \times \left(\sum (y_{i-\bar{y}})^2\right)}} \tag{1}$$

where *r* is the Pearson correlation coefficient of a group of variables *x* with another group of variables *y*.

In turn, a multiple linear regression analysis was performed. The dependent variable is annual coastline retreat, and the independent variables are annual dredging, the total length of groins, and annual precipitation. The goal of multiple linear regression analysis is to find a regression equation that provides a better perception of the sign and the extent to

which independent variables determine values of the dependent variable—thus, aiming to find regression coefficients $\beta$ that best fit the dependent variable $y$ [35]:

$$y = \beta_0 + \beta_1 x_1 + \ldots + \beta_n x_n + \varepsilon \tag{2}$$

where $y$ is the dependent variable, $x$ are the independent variables, $\beta$ are the regression coefficients, and $\varepsilon$ is the residue error of prediction. The latter is the difference between the actual values and those predicted by the regression model and is assumed to be normally distributed with a zero mean and constant variance ($\sigma^2$) [34,35].

In seeking to determine the extent to which anthropogenic and natural factors explain coastline retreat in Vagueira Beach, the following multiple linear regression model was developed:

$$Retreat = \beta_0 + \beta_1 Groin5 + \beta_2 Dred3 + \beta_3 Precp \tag{3}$$

where *Retreat* is the annual coastline retreat in Vagueira Beach (in m/year), *Groin5* is the total length of groins built for the Espinho–Vagueira section at $t + 5$ (in m), *Dred3* is the annual dredging volume at the Aveiro Port entrance at $t + 3$ (in 100,000 m³/year), and where *Precp* is the annual precipitation (in mm/year). A time lag of $t + 5$ years for the total length of groins and $t + 3$ years for the dredging volume was adopted, as the lack of sediments in Vagueira Beach is noticed in about this time scale [16,17,22,25].

Finally, the estimated regression model was validated using Student's *t*-test, the determination coefficient ($R^2$), the ANOVA *F*-test, and the variance inflation factor (*VIF*). Student's *t*-test was used to check whether the signs and magnitude of the regression coefficients made sense in the context of the phenomenon being studied [35]. Furthermore, tests and confidence intervals (*t*-test and *F*-test) allow an indirect idea of the regression quality to be obtained and confirm the hypotheses of particular values for the parameters established by theoretical means.

The determination coefficient ($R^2$) represents a measure of adjustment for a statistical model in relation to the observed values, i.e., it indicates, in percentage terms, the extent to which the regression model explains the observed values. The closer $R^2$ is to 1, the larger the explanatory power or fit of the model [35].

The ANOVA *F*-test allows us to evaluate the overall model, which, in general, is the statistical testing of the confidence of the coefficient of determination. Thus, if the *F*-test indicates a low level of significance (less than 5%), the data estimated by the regression model are close to those of the initially observed data.

Multicollinearity occurs when independent variables are strongly correlated, so that the interpretation of the contribution of predictors becomes difficult, and the estimation of the regression coefficients is flawed. To test whether this occurs, the variance inflation factor (*VIF*) was calculated. A *VIF* > 10 indicates that the regression model shows problems of multicollinearity. The *VIF* is given by [36]:

$$VIF = \frac{1}{1 - R^2} \tag{4}$$

where *VIF* is the variance inflation factor of the regression model with a coefficient of determination $R^2$.

## 3. Results

### 3.1. Analysis of Correlation between the Factors Related to Coastline Retreat

The retreat rate of the coastline is one of the best indexes of coastal erosion. It is a unique index and of exceptional value for the evaluation of coastline evolution tendencies as well as for the assessment of the real impacts of disruptive factors on coastal sediment dynamics [25]. In order to assess the relationship between the factors that can contribute to coastal erosion, correlation analysis was used to correlate the annual coastline retreat at Vagueira Beach with variables that are considered to influence coastal erosion—namely

(Table 2): annual dredging, the total length of groins, and annual precipitation (as a proxy for storm events).

**Table 2.** Correlation analysis between coastline retreat and possible factors that influence coastal erosion in Vagueira Beach.

| | Coastline Retreat in Vagueira Beach | |
| --- | --- | --- |
| **Factors** | **Pearson's Correlation** | **N° of Observations (Evaluated years)** |
| Dredging $t + 3$ | 0.931 *** | 27 |
| Total length of groins $t + 5$ | 0.891 *** | 27 |
| Precipitation | 0.521 *** | 22 |

Significant at the 1% (***) level.

A high correlation was obtained between coastline retreat and the dredging at the Aveiro Port entrance (0.931), even though the port entrance is located around 10 km north of Vagueira Beach. The sediment dredged from the port entrance creates a deficit of sediment along the coastal stretch to the south. This lack of sediments is noticed in Vagueira Beach about 3 years after dredging ($t + 3$) at the Aveiro Port entrance (see Figure 5).

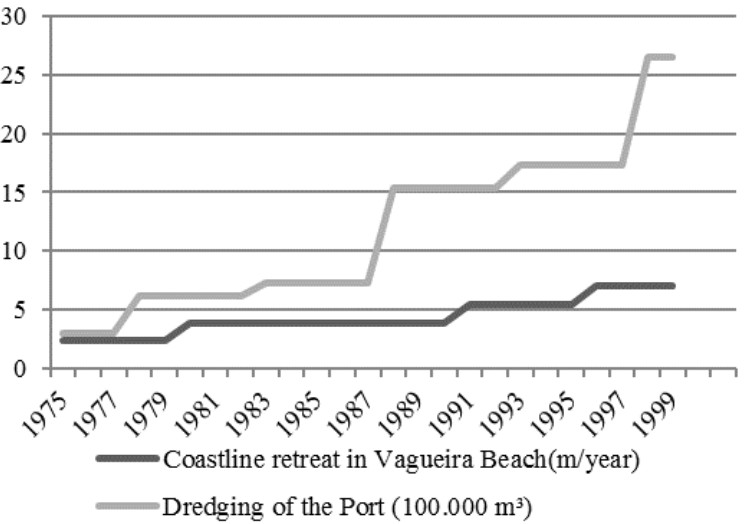

**Figure 5.** Cumulative annual volume dredged at the Aveiro Port entrance and cumulative annual coastline retreat at Vagueira Beach over the period 1975–2000 (based on: [7]).

The total length of groins also presents a high correlation with coastline retreat (0.891). Coastal protection measures are generally established in front and just south of urbanized areas [16]. As with the dredging process, groins reduce sediment transport along the coast, which, according to the correlation analysis, takes 5 years to be transported to Vagueira Beach ($t + 5$).

Finally, the coastal area is influenced by oceanic, continental, and atmospheric agents, hence its sensitivity to climate conditions [37]. Even though the correlation between coastline retreat and precipitation is relatively low (0.521), it seems that annual precipitation (and associated storm events) is a reasonable proxy for the energetic actions of the sea (see Figure 6). These actions have generated events of great impact that could jeopardize works of containment and coastal defense—leading to the relocation of sediments confined by coastal defense structures [22]. Such relocation of sediments would, however, not result in recovery because there is a net sediment deficit (i.e., accretion is smaller than erosion; see Figure 2). In addition, although groins were sporadically and locally damaged over the years, groins are hard coastal structures that have been maintained over time and, thus, have preserved their full performance.

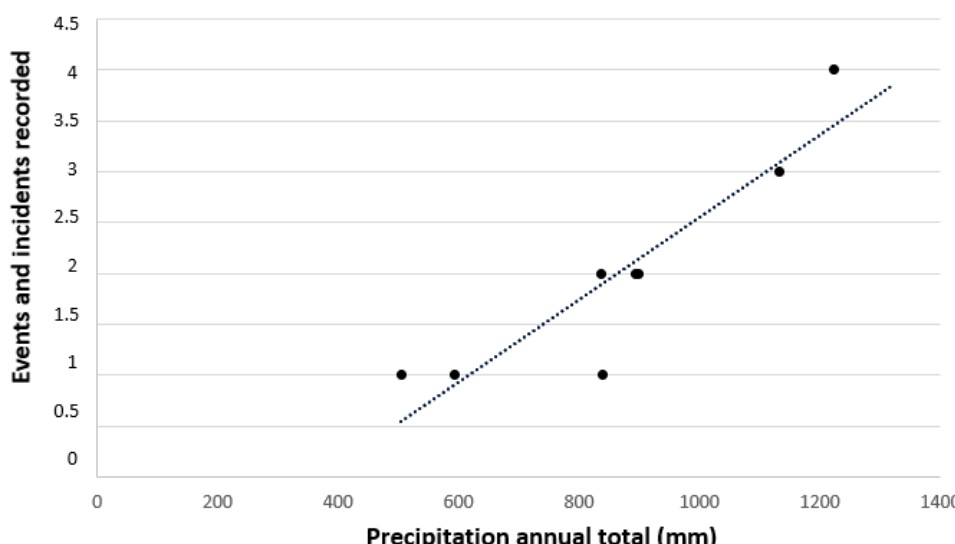

**Figure 6.** Annual precipitation and number of recorded events and incidents of wave actions in Vagueira Beach (based on: [22,32]).

*3.2. Assessment of Factors Related to Coastal Erosion through Multiple Linear Regression*

Regression simulation results (see Table 3) show that the model has a very good fit ($R^2 = 0.91$). The ANOVA *F*-test (significant at 1%) indicates that values estimated by the regression model are close to the initial data of annual coastline retreat in Vagueira Beach, while the variance inflation factor (*VIF*) indicates that there is no problem of multicollinearity (*VIF* < 10).

**Table 3.** Results for the multiple linear regression model of coastline retreat in Vagueira Beach.

| Explainatory Variables | Coefficients | | Student *t*-Test | *VIF* |
|---|---|---|---|---|
| | Estimate | Weight | | |
| Constant ($\beta_0$) | 0.615 | | 0.912 | |
| *Groin5* ($\beta_1$) | 0.002 | 0.526 | 3.074 *** | 5.773 |
| *Dred3* ($\beta_2$) | 0.009 | 0.384 | 2.171 ** | 6.165 |
| *Precp* ($\beta_3$) | 0.001 | 0.130 | 1.611 * | 1.280 |
| $R^2$ | | | 0.91 | |
| ANOVA *F*-test | | | 59.79 *** | |

Significant at the 1% (***), 5% (**) and 10% (*) level.

The analysis of the factors that explain annual coastline retreat allows for a series of interpretations (Table 3). All factors have a positive sign, indicating that they are directly proportional to coastline retreat. The contribution of the total length of the groins is the main factor responsible for coastline retreat in Vagueira Beach (*Groin5* with a weight of 53%), followed by the volume dredged at the Aveiro Port entrance (*Dred3* with a weight of 38%), and precipitation (*Precp* with a weight of 13%). Several authors argue that the weakening of sedimentary sources in the region is the biggest cause of coastal erosion problems [16,17,22,25]. The reduction in sedimentary sources can explain the rates of coastline retreat observed in the region. This is in line with the obtained results, which show that groins and dredging reduce the transport of sediments and that over the years (after 5 and 3 years, respectively), this would settle on the coast in front of Vagueira Beach [22].

Student's *t*-test reveals that the total length of groins (*Groin5*) is the most significant factor (significant at the 1% level), followed by the volume dredged at the Aveiro Port entrance (*Dred3*; significant at the 5% level) proving, once again, the importance of sediment removal in explaining coastline retreat and confirming that human interventions lead to

environmental changes. Finally, precipitation (*Precp*) is the least significant factor explaining coastline retreat (significant at the 10% level), indicating that annual precipitation, and associated storm events, only have a limited impact on shoreline retreat—albeit storm events cause significant damages to coastal infrastructures (see, e.g., [22,25,31]).

## 4. Discussion

Coastal erosion is one of the major management problems that coastal regions face worldwide [38], and accurate information on coastline movement rates and trends is essential to support sustainable management strategies. Coastline retreat is a dominant trend along the Portuguese coastal zone, with a mean rate of retreat of 0.24 ± 0.01 m per year for the mainland [8]. The central Portuguese coast, where Vagueira Beach is located, has suffered severe coastal erosion for decades—particularly along the sandy stretches. This process is related to anthropogenic transformations, a reduction in the supply of sediments, an increase in the mean sea level, and an increase in storm events [23].

The construction of coastal defense structures in Vagueira Beach began at the end of the 1970s and was accompanied by an increase in urban occupation [23]. Hence, the density of the built area has increased significantly over the past decades—with the urban front increasing to 650 m in 2015. Along the coastline under analysis, coastal erosion has been addressed through the construction of protection structures, mainly groins, and dikes, which have had a negative impact on (beach) tourist demand. However, as the coastline retreat continues, it is expected that additional defense construction, maintenance, and emergency interventions will be needed with corresponding financial implications.

In the past, the sediments coming from continental sources, mainly rivers, would have the capacity to practically alone supply the sediments necessary for equilibrium [11]. However, interventions such as the construction of dams in rivers north of Vagueira Beach [8,22], dredging at the Aveiro Port entrance [22,29], and the construction of coastal defense structures [23,29], pinpoint the underlying problem: the lack of sediment and subsequent coastal erosion. Several authors (see, e.g., [8,9,21,23,39]) are unanimous in stating that the weakening of sedimentary sources is the major cause of erosion problems in the area under analysis [18].

Climate change generates several impacts on the coastal zone, such as the potential rise in sea levels that puts further pressure on shoreline retreat (see, e.g., [10,12,40] as well as the increase in storm events that is expected to result in higher and also the altered direction of waves (see, e.g., [16,25]). As for precipitation, periods of drought alternating with periods of heavy precipitation, can lead to a decrease in river flows and, thus, less export of sediments by rivers to the coast (see, e.g., [22,40]).

## 5. Conclusions and Recommendations

This study assessed the influence of anthropogenic and natural factors on coastline retreat in Vagueira Beach (Central Portugal). To this end, correlation and multiple linear regression analyses were applied. The main variables correlated with coastline retreat in Vagueira Beach were the total length of groins, annual dredging, and, to a minor extent, annual precipitation (as a proxy for storm events). These results are in line with other authors that highlighted the influence of sediment deficits on downdrift coastline retreat [22] associated with human interventions (such as dredging) and coastal structures (such as groins) [29] as well as the influence of climatic factors [16,25].

Through the temporal distribution of several factors in the analysis, it can be concluded that the coastline retreat in Vagueira Beach has increased and is expected to maintain this trend in the near future if no other mitigation measures are considered. In particular, this is observed because of the groins constructed along the coastal stretch north of Vagueira Beach and the volume of material dredged at the Aveiro Port entrance, which has shown an increasing trend [22]. In addition, climate change is expected to lead to sea-level rises and an increase in storm events over the next century [14,21,25]. Hence, the construction of groins along the coastal stretch north of Vagueira Beach should be carefully deliberated,

while the recharge of materials dredged at the Aveiro Port entrance to beaches located south is increasingly being considered to mitigate coastal erosion on this coastal stretch [21–25]. In the meantime, local solutions in front of Vagueira Beach are assessed to halt and reverse coastline retreat in Vagueira Beach [10,13,18,37].

Some caveats remain. First, the use of annual precipitation as a proxy for storm events and associated wave actions has proven to work only reasonably well. A complete time series of actual wave action would be preferred, though the maritime wave records were too incomplete to be used for this purpose. Second, river sediment recharge from the main river to the north of Vagueira Beach (in particular, from the Vouga River) would be an additional explanatory variable of importance, albeit most of the sediments from the Vouga River are trapped in the Aveiro Lagoon and Aveiro Port entrance. Finally, unlike other studies in this area, which focus on monitoring and modeling, this study provided an innovative statistical approach, based on correlation and regression analysis, to assess the causes and impacts of coastal erosion and can be easily replicated in similar coastal zones.

**Author Contributions:** Conceptualization, L.M.F., P.R. and C.C.; data curation, L.M.F., P.R. and C.C.; formal analysis, L.M.F. and P.R.; investigation, L.M.F., P.R. and C.C.; methodology, L.M.F. and P.R.; supervision, P.R.; validation, C.C. and P.R.; writing—original draft, L.M.F., P.R. and C.C.; writing—review and editing, L.M.F., P.R. and C.C. All authors have read and agreed to the published version of the manuscript.

**Funding:** This study was financed in part by the Coordenação de Aperfeiçoamento de Pessoal de Nível Superior—Brasil (CAPES)—Finance Code 001 and acknowledges the support from the ECOMAR project funded by the CYTED program. In addition, this work was financially supported by the project "Integrated Coastal Climate Change Adaptation for Resilient Communities", INCCA—POCI-01-0145-FEDER-030842, funded by FEDER, through "Competividade e Internacionalização" in its FEDER/FNR component and by national funds (OE), through FCT/MCTES. Finally, thanks are due for the financial support of FCT/MCTES and for the financial support of CESAM (UIDB/50017/2020 and UIDP/50017/2020) through national funds and the co-funding by European funds when applicable.

**Institutional Review Board Statement:** Not applicable.

**Informed Consent Statement:** Not applicable.

**Data Availability Statement:** Not applicable.

**Acknowledgments:** Thanks are due for the financial support of the Coordenação de Aperfeiçoamento de Pessoal de Nível Superior—Brasil (CAPES)—Finance Code 001 and acknowledges the support from the ECOMAR project funded by the CYTED program. In addition, this work was financially supported by the project "Integrated Coastal Climate Change Adaptation for Resilient Communities", INCCA—POCI-01-0145-FEDER-030842, funded by FEDER, through "Competividade e Internacionalização" in its FEDER/FNR component and by national funds (OE), through FCT/MCTES. Finally, thanks are due for the financial support of FCT/MCTES and for the financial support of CESAM (UIDB/50017/2020 and UIDP/50017/2020) through national funds and the co-funding by European funds when applicable.

**Conflicts of Interest:** The authors declare no conflict of interest.

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
