# Peer review of "Influence of Local Factors on Coastal Erosion: The Case of Vagueira Beach in Portugal"

_environments, doi:10.3390/environments10020024_

Round 1

Reviewer 1 Report

This paper describes coastal erosion on Vagueira Beach, the Central Portuguese coast. This paper assesses the anthropogenic and natural factors that are related with the retreat of the coastline, by adopting statistical correlation and regression analyses. The paper has practical importance related to local influential factors causing beach erosion. In order to be published in the journal, however, further descriptions and explanations are highly required as listed below. Especially, the interpretation for each factor is unsatisfactory, only with qualitative descriptions. My comments are mainly from this view point.

(1) Regarding the influence of the groin, the authors used Figure 2 to explain the erosion process in the downdrift area. As seen in the figure, however, it is well known that on the updrift side, considerable sediment deposit can be observed. Therefore, about the influence the groin, it is totally one sided in this MS. More detailed discussion is required for what happened on the updrift side of the grain.

(2) About precipitation, Figure 6 clearly indicates that wave height is much more suitable as a parameter to be used in the present regression analysis. Precipitation is too unsuitable as an explanatory variable in this kind of study in the field of coastal and ocean engineering.

(3) P.7 at the bottom: If the groin is destroyed by violent waves, sediment can bypass through the damaged coastal structure, which might result in recovery of the shoreline? Like this both sided interpretations can be made for the influence of precipitation, since the discussion is highly qualitative.

(4) According to Figure 3, the length of the groin has already reached the final length. Then, very big groin effect will exist here forever in this coastal area? No stable situation anymore, even if no dredging and normal precipitation condition? Such a future projection cannot be made using the present regression model? Although this multiple regression analysis has been made successfully to some extent, this cannot be utilized for future projection, as it is lacking discussions based on associated physical processes.

(5) It is well known that general sediment movement can be classified into longshore and cross-shore movements. It is necessary to elaborate which one is predominant due to three factors handled in this study.

Author Response

Response to Reviewers Comments

Article: Influence of local factors on coastal erosion: the case of Vagueira beach in Portugal

Reference: environments-2072509

Journal: Environments

Response to Editors:

Dear Editors,

Please find attached the revised submission and response to reviewers of the paper Influence of local factors on coastal erosion: the case of Vagueira beach in Portugal, to be considered for publication in Environments. We thoroughly revised the paper, according to suggestions given by the reviewers and the Editor in charge of the manuscript. We are thankful for the constructive comments and suggestions, which has contributed to the quality of the manuscript. Our detailed response to the reviewers’ comments is provided below.

Reviewer 1

This paper describes coastal erosion on Vagueira Beach, the Central Portuguese coast. This paper assesses the anthropogenic and natural factors that are related with the retreat of the coastline, by adopting statistical correlation and regression analyses. The paper has practical importance related to local influential factors causing beach erosion. In order to be published in the journal, however, further descriptions and explanations are highly required as listed below. Especially, the interpretation for each factor is unsatisfactory, only with qualitative descriptions. My comments are mainly from this view point.

Response to Reviewer: We thank the reviewer for the constructive comments and suggestions, based on which we thoroughly revised the paper (see responses to your comments and suggestions below). This greatly improved the quality of the manuscript. Albeit we believe we adequately addressed the issues raised and corrections mentioned by the reviewer, please do not hesitate to contact us for any remaining issues.

(1) Regarding the influence of the groin, the authors used Figure 2 to explain the erosion process in the downdrift area. As seen in the figure, however, it is well known that on the updrift side, considerable sediment deposit can be observed. Therefore, about the influence the groin, it is totally one sided in this MS. More detailed discussion is required for what happened on the updrift side of the groin.

Response 1: Groins are short structures placed perpendicular to the coastline and extended to the surf zone, usually built-in straight portions of the shoreline. Their main purpose is to retain sand and promote updrift accretion (Bush et al., 2001). Groins do not add sediment to the coastal system and secondary negative effects of these structures may anticipate sediment deficit downdrift. At Vagueira Beach, the main negative impacts are registered due to the northern structures, and this is explained by a predominance of currents from the north-west along the considered coastal stretch (Pedrosa, 2013). This notion has now been added before figure 2 of Section 2 (Materials and Methods).

(2) About precipitation, Figure 6 clearly indicates that wave height is much more suitable as a parameter to be used in the present regression analysis. Precipitation is too unsuitable as an explanatory variable in this kind of study in the field of coastal and ocean engineering.

Response 2: We fully agree that wave height is a better explanatory variable than precipitation (as a proxy for storm events). However, wave records during storm events are unreliable (incomplete), as during these events measuring equipment frequently fails due to the high wave energy. Hence, we opted to use precipitation as a proxy for storm events, given that periods of intense rainfall (mainly during autumn and winter) are, generally, accompanied by storm events. This is explained in Section 2.1 (Characterization of the study area) and confirmed by Figure 6.

(3) P.7 at the bottom: If the groin is destroyed by violent waves, sediment can bypass through the damaged coastal structure, which might result in recovery of the shoreline? Like this both sided interpretations can be made for the influence of precipitation, since the discussion is highly qualitative.

Response 3: Storm events could, indeed, jeopardize works of containment and coastal defense – leading to the relocation of sediments confined by coastal defense structures. Such re-location of sediments would, however, not result in recovery because there is a net sediment deficit (i.e. accretion is smaller than erosion). In addition, albeit groins were sporadically and locally damaged over the years, groins are hard coastal structures that were maintained over time and, thus, preserved their full performance. This notion has now been added in Section 3.1 (Analysis of correlation between the factors related to coastline retreat), before figure 6.

(4) According to Figure 3, the length of the groin has already reached the final length. Then, very big groin effect will exist here forever in this coastal area? No stable situation anymore, even if no dredging and normal precipitation condition? Such a future projection cannot be made using the present regression model? Although this multiple regression analysis has been made successfully to some extent, this cannot be utilized for future projection, as it is lacking discussions based on associated physical processes.

Response 4: Note that groin sand retention and accretion updrift and, in particular, sand depletion and erosion downdrift takes several decades before a stable situation is attained (Bush et al., 2001) and, hence, impacts of groins are observed until decades after their construction.

(5) It is well known that general sediment movement can be classified into longshore and cross-shore movements. It is necessary to elaborate which one is predominant due to three factors handled in this study.

Response 5: A more detailed description of longshore and cross-shore sediment movements along the considered coastal stretch has been added after Table 1 and before figure 2, in Section 2 (Materials and Methods).

Reviewer 2 Report

This work is well-written and considers an important study for the assessment of coastal erosion. In particular, the sandy coast of Vaguiera Beach (Central Portugal) was studied considering the anthropogenic structures and their influence for the coastal erosion. I have just few comments to improve this manuscript. I suggest to divide the subsection 2.1 in another new section called "Geographic framework". Then other few comments are highlighted in the attached pdf file.

Many thanks and kind regards.

Author Response

Response to Reviewers Comments

Article: Influence of local factors on coastal erosion: the case of Vagueira beach in Portugal

Reference: environments-2072509

Journal: Environments

Response to Editors:

Dear Editors,

Please find attached the revised submission and response to reviewers of the paper Influence of local factors on coastal erosion: the case of Vagueira beach in Portugal, to be considered for publication in Environments. We thoroughly revised the paper, according to suggestions given by the reviewers and the Editor in charge of the manuscript. We are thankful for the constructive comments and suggestions, which has contributed to the quality of the manuscript. Our detailed response to the reviewers’ comments is provided below.

Reviewer 2

This work is well-written and considers an important study for the assessment of coastal erosion. In particular, the sandy coast of Vaguiera Beach (Central Portugal) was studied considering the anthropogenic structures and their influence for the coastal erosion. I have just few comments to improve this manuscript. I suggest to divide the subsection 2.1 in another new section called "Geographic framework". Then other few comments are highlighted in the attached pdf file.

Response to Reviewer: Thank you for the detailed comments in the .pdf file, which we by-and-large have adopted accordingly. About the division in the subsection 2.1 in another new section called "Geographic framework", we have given your suggestions consideration, though based on feedback from the other reviewers we decided to maintain the current structure. About Table 1, the rates of coastline retreat used in this study were obtained from EUROSION (EU, 2022) though, unfortunately, do not include standard deviations. A good paper in this respect is Lira et al. (2016), which provides such information for large coastal cells along the Portuguese continental coast. This paper is now referenced in the paper.

Reviewer 3 Report

The article is interesting, and the subject is worthy of research. I find it really well-thought-of and well-written. However, the execution of the article requires some improvements to proceed with its publication in the journal. For this, I advise a major revision to the authors in the following points.

In the introduction, the authors miss a number of recent articles that deals with coastal erosion analysis. Citing 21 articles seems quite good but most of them are articles from the years 2005-2013 and the newest citing is from 2015. Moreover, a significant percentage of those works are self-citations. This suggests that the literature review has to be redone. Numerous new works and analyses are available. Plenty of these use also sophisticated statistical methodologies such as Bayesian networks, machine learning techniques (both classifiers and regression tools), and others for erosion analysis of both dune and cliff coasts.  Please update the introduction with new positions which could later be also used for more extended discussion.

I really like the methodology and result sections. What I miss is a more detailed description of acquired data. Most data has been obtained from projects or partners. In section 2.2. authors address those with references. Regardless of this, it would be helpful to add a description. Now the reader has to search through different documents to find information about data collection protocols, etc.

Finally what I miss is a discussion section. Authors from the results jump directly to conclusions. Actually, this section seems as a semblance of discussion. In my opinion, it should be presented as a separate section. Please keep in mind to compare the methodology and results with different studies more widely not only with authors' earlier articles. Having the Introduction extended it would be reasonable to refer to those new works.

Additional minor changes:

While presenting maps please always add scale and a north arrow.

Author Response

Response to Reviewers Comments

Article: Influence of local factors on coastal erosion: the case of Vagueira beach in Portugal

Reference: environments-2072509

Journal: Environments

Response to Editors:

Dear Editors,

Please find attached the revised submission and response to reviewers of the paper Influence of local factors on coastal erosion: the case of Vagueira beach in Portugal, to be considered for publication in Environments. We thoroughly revised the paper, according to suggestions given by the reviewers and the Editor in charge of the manuscript. We are thankful for the constructive comments and suggestions, which has contributed to the quality of the manuscript. Our detailed response to the reviewers’ comments is provided below.

Reviewer 3

The article is interesting, and the subject is worthy of research. I find it really well-thought-of and well-written. However, the execution of the article requires some improvements to proceed with its publication in the journal. For this, I advise a major revision to the authors in the following points.

Response to Reviewer: We thank the reviewer for the constructive comments and suggestions, based on which we thoroughly revised the paper (see responses to your comments and suggestions below). This greatly improved the quality of the manuscript. Albeit we believe we adequately addressed the issues raised and corrections mentioned by the reviewer, please do not hesitate to contact us for any remaining issues.

(1) In the introduction, the authors miss a number of recent articles that deals with coastal erosion analysis. Citing 21 articles seems quite good but most of them are articles from the years 2005-2013 and the newest citing is from 2015. Moreover, a significant percentage of those works are self-citations. This suggests that the literature review has to be redone. Numerous new works and analyses are available. Plenty of these use also sophisticated statistical methodologies such as Bayesian networks, machine learning techniques (both classifiers and regression tools), and others for erosion analysis of both dune and cliff coasts.  Please update the introduction with new positions which could later be also used for more extended discussion.

Response 1: Additional and more recent references have been added to the Introduction, Discussion and Conclusions sections, namely:

Bush, D.; Pilkey, O.; Neal, W. Human impact on coastal topography. Ency. of Ocean Sciences (Second Edition). 2001. 581-590. doi: 10.1016/B978-012374473-9.00078-3.    

Lira, C.; Nobre Silva, A.; Taborda, R.; Freire de Andrade, C. Coastline Evolution of Portuguese Low-Lying Sandy Coast in the Last 50 Years: an Integrated Approach. Earth Syst. Sci. Data, 2016. 8, 265-278. doi: 10.5194/essd-8-265-2016.

Maia, A.; Bernardes, C.; Alves, M. Cost-benefit analysis of coastal defenses on the Vagueira and Labrego beaches in North West Portugal. J. of Inte. Coast. Zone Man., 2015. 15(1), 81-90. doi:10.5894/rgci521. 

Phillips, M.; Jones, A. Erosion and tourism infrastructure in the coastal zone: Problems, consequences and management. Tour. Manag. 2006. 27, 517–524. doi:10.1016/j.tourman.2005.10.019.

Pinto, C.; Silveira, T.; Teixeira, S. Beach nourishment practice in mainland Portugal (1950–2017): Overview and retrospective. Ocean & Coastal Man., 2020.  192, 105211. doi: 10.1016/j.ocecoaman.2020.105211.

Santos, F.; Lopes, A.; Moniz, G.; Taborda, R.; Ramos, V. Gestão da Zona Costeira: O Desafio da Mudança. Portuguese Littoral Working Group Report, 2014. 237 p. (in Portuguese). Available online: https://apambiente.pt/sites/default/files/_SNIAMB_Agua/DLPC/ENGIZC/GTL_RF20150416.pdf.

Stefanova, A.; Krysanova, V.; Hesse C.; Lillebø, A. Climate change impact assessment on water inflow to a coastal lagoon: the Ria de Aveiro watershed, Portugal. Hydro. Science J. 2015. 60 (5). doi: 10.1080/02626667.2014.983518.

Veloso-Gomes, F.  Contributions for the Revision of the Coastal Zone Management of the Portuguese Central Region. 5th. Journeys of Hydraulics, Water Resources and Environment. 2010. 7 p. (in Portuguese). Available online: https://paginas.fe.up.pt/~shrha/publicacoes/pdf/JHRHA_5as/3_VGomes_ContributosCentro.pdf.

(2) I really like the methodology and result sections. What I miss is a more detailed description of acquired data. Most data has been obtained from projects or partners. In section 2.2. authors address those with references. Regardless of this, it would be helpful to add a description. Now the reader has to search through different documents to find information about data collection protocols, etc.

Response 2: Section 2.2 (Data collection) details the different data sources used and that were described in the previous section. We believe that the journal "Environments" uses hyperlinks in their references and, thus, allows the reader to directly access the mentioned sources.

(3) Finally what I miss is a discussion section. Authors from the results jump directly to conclusions. Actually, this section seems as a semblance of discussion. In my opinion, it should be presented as a separate section. Please keep in mind to compare the methodology and results with different studies more widely not only with authors' earlier articles. Having the Introduction extended it would be reasonable to refer to those new works.

Response 3: A Discussion section has now been included (see Section 4), providing a comparison with previous studies in this area. We have added several, non-self-citing, studies.

(4) Additional minor changes:

While presenting maps please always add scale and a north arrow.

Response 4: Figure 1 changed, including scales and a north arrows.

Round 2

Reviewer 1 Report

None

Reviewer 3 Report

Congratulation to the authors. I accept the introduced changes.